# Recent Methods of Kidney Storage and Therapeutic Possibilities of Transplant Kidney

**DOI:** 10.3390/biomedicines10051013

**Published:** 2022-04-28

**Authors:** Anna Radajewska, Anna Krzywonos-Zawadzka, Iwona Bil-Lula

**Affiliations:** Department of Medical Laboratory Diagnostics, Division of Clinical Chemistry and Laboratory Hematology, Faculty of Pharmacy, Wroclaw Medical University, Borowska 211A, 50-556 Wroclaw, Poland; anna.radajewska@student.umw.edu.pl (A.R.); anna.krzywonos-zawadzka@umw.edu.pl (A.K.-Z.)

**Keywords:** machine perfusion, kidney transplantation, kidney graft

## Abstract

Kidney transplantation is the standard procedure for the treatment of end-stage renal disease (ESRD). During kidney storage and before implantation, the organ is exposed to damaging factors which affect the decline in condition. The arrest of blood circulation results in oxygen and nutrient deficiency that lead to changes in the cell metabolism from aerobic to anaerobic, damaging organelles and cell structures. Currently, most kidney grafts are kept in a cold preservation solution to preserve low metabolism. However, there are numerous reports that machine perfusion is a better solution for organ preservation before surgery. The superiority of machine perfusion was proved in the case of marginal donor grafts, such as extended criteria donors (ECD) and donation after circulatory death (DCD). Different variant of kidney machine perfusions are evaluated. Investigators look for optimal conditions to protect kidneys from ischemia-reperfusion damage consequences by examining the best temperature conditions and comparing systems with constant or pulsatile flow. Moreover, machine perfusion brings additional advantages in clinical practice. Unlike cold static storage, machine perfusion allows the monitoring of the parameters of organ function, which gives a real possibility to make a decision prior to transplantation concerning whether the kidney is suitable for implantation. Moreover, new pharmacological therapies are sought to minimize organ damage. New components or cellular therapies can be applied, since perfusion solution flows through the organ. This review outlines the pros and cons of each machine perfusion technique and summarizes the latest achievements in the context of kidney transplantation using machine perfusion systems.

## 1. Introduction

Chronic kidney disease (CKD) might lead to end-stage renal disease. In 2017, 1.2 million people died because of chronic kidney disease [1]. There are two therapeutic options for ESRD, namely dialysis and transplantation. The development of organ preservation is indisputably a big advantage of medicine and transplantology. The main goal of the customarily used method (hypothermia) is to suppress the metabolism and reduce ATP depletion. Increasing the pool of organs for transplantation is still an important objective because of the long waiting list for organ transplants [2]. With the increase in the number of patients qualified for transplantation, scientists are forced to find a way to increase the organ pool. Currently, it is possible to obtain a kidney from a living donor, after brain death and after cardiac death. There is also a group of donors called extended criteria donors that are approved to be organ donors. This change was a big step for transplantology, but it is also a challenge for clinicians and scientists to ensure the best function of the organ. The composition of a storage fluid and the preservation method seem to be crucial. One of the first preservation solutions used for cold static storage was plasma and later, cryoprecipitate plasma (CPP). The problem with the possibility of disease transmission was the reason for the production of a commercial product supporting organ preservation. The construction of machine perfusion devices and the development of a constant perfusion method were significant in the case of donation after cardiovascular determination of death. It is known that DCD organs are more prone to ischemia-reperfusion injury (IRI). IRI is the main cause of delayed graft function (DGF) or primary nonfunction (PNF) of the graft. That is why a new method of preservation, in combination with solution composition, should be considered [3,4,5,6]. According to the study of Opelz et al., during the years 1995–2005, machine perfusion constituted only 2.0–3.4% of all preserved kidneys [5]. Fortunately, the popularity of machine perfusion is increasing, so we can expect that it will completely supersede cold static storage (CSC) of the kidney [7]. The standardization of machine perfusion methods and the improvement of perfusing solutions should be the next step.

## 2. What Occurs in the Kidney during the Storage?

After organ removal from a donor’s body and before transplantation, a graft needs to be stored for different periods of time. The duration of organ preservation is proportional to the time of ischemia and the degree of the damage that occurs. Ischemia results in oxygen and nutrition deficiency and promotes cell apoptosis or necrosis. Inside the cell, numerous biochemical changes occur. Anaerobic respiration is promoted, which leads to adenosine triphosphate (ATP) depletion and inactivation of the Na^+^/K^+^ ATPase pump. ATP is rapidly transformed into hypoxanthine, which can be subsequently metabolized because of anaerobic conditions (Figure 1). The first phase of ATP depletion is associated with the inactivation of manganese superoxide dismutase (SOD). Because of the inhibition of its antioxidative properties, mitochondria are destroyed by reactive oxygen species (ROS), and some apoptotic pathways are launched. Moreover, lactate dependent ATP production leads to acidosis because of lactic acid accumulation inside the cell and interstitium. These conditions affect lysosome membranes and hydrolase leaking, destroying cell membrane structure [8,9]. Because of the Na^+^/K^+^ ATPase failure, increasing intracellular Na^+^ ion concentration and water accumulation yield edema. Further, Ca^2+^ accumulation is caused by Na^+^/Ca^2+^ pump arrest. High amounts of calcium activate calcium-dependent proteases, such as calpains (active after reperfusion), and initiate reactive oxygen species production [8,10]. These changes are proportional to the time of ischemia duration. Due to IR, injured cells activate different types of cell death: apoptosis, necrosis or autophagy. Apoptosis is determined by caspase complex activation and can be classified as a self-limiting death program. It was thought that apoptosis is less immunogenic than necrosis, but ATP release might attract phagocytes and initiate inflammation. Necrosis, on the other hand, is initiated by cellular swelling and membrane breakage. Injured cells stimulate the immune system, infiltration of the neutrophils and a cytokine cascade reaction. Autophagy is a genetically regulated process in which cells start to digest organelles and intracellular proteins inside their own lysosomes, which can be considered as a temporary survival mechanism. However, prolonged ischemia time might result in the cellular death process rather than in survival action [10]. When the blood flow is restored, white blood cells release cytokines and chemoattractant, which injures endothelial and tubular epithelial cells [9]. During the immune response, the activation of both pathways of the innate and adaptive immune system are observed. The transcription of nuclear factor kinase B, by activating toll-like receptors, leads to the transcriptional activation of inflammatory genes. Dendritic cells have a crucial role in immune response. They present an antigen to T cells and might activate the complement system. Usually, T cells are activated by binding with antigens; however, it has been shown that they might be activated by ROS. In the pathophysiology of the immune system response in the case of kidney transplantation, CD4^+^ lymphocytes are of particular importance. During cold preservation, there is increased expression of P-selectin, ICAM-1 and MHC class II antigen on renal endothelial and proximal tubular cells before CD4^+^ T-cell infiltration. Increased expression of the MHC class II antigen is associated with release of interleukin 2 (IL-2) and interferon gamma (INF-γ). This suggests an innate connection and an adaptive immune system in the immune reaction [11].

## 3. Expanded Criteria in Kidney Graft Transplantation

There is still increasing disproportion between organs available for transplantation and the number of waiting recipients (Table 1) [12,13]. Previously, kidney transplants were carried out after the donor’s brain stem death. Currently, neurosurgery is more effective, and we can observe the reduction of postsurgical trauma. This fact caused the need to expand the criteria for organs that can be used [12]. Eventually, new criteria for donors were approved and included ECD and DCD. For the ECD group, one can obtain grafts from deceased donors over sixty, or 50–60-year-old donors with two other criteria: arterial hypertension, creatine levels over 1.5 mg/dL and DCD grafts [14]. The overriding standard is the organ’s condition. This means that broad structural lesions, functional disorders or active infection of the organ does not allow it to be used during transplantation [15]. DGF is a common condition occurring after renal transplantation. It enlarges the risk of transplant rejection in the future and requires the implementation of hemodialysis, which also increases the total cost and affects the patient’s quality of life [16]. Studies have shown that DCD grafts are associated with a higher risk of DGF compared to kidneys recovered after brain death donation (DBD), likely because of extended warm ischemic time [17], although a DCD kidney is less likely to affect the hormonal and inflammatory milieu than a DBD kidney [18]. However, in both cases, kidneys are exposed to ischemia during preservation, causing injuries that might be an important factor leading to DGF or graft rejection. This necessitates the improvement of graft storage to stop destructive processes from occurring during cold ischemia time [19].

## 4. Preservation Solutions

At the beginning of transplantology development, researchers and clinicians were focused mainly on the improvement of surgical techniques, immunosuppression and finally, appropriate storage and obtaining the best preservation solution [19,20,21]. In 1969, Collins et al. obtained a preservation solution that was able to maintain the cold storage of kidney grafts for up to 30 h [22]. A few years later, the solution was modified to better protect the graft from prolonged ischemia injury. The new solution was called Euro-Collins. In the 1980s, the University of Wisconsin (UW) solution was released to the market, and it is still used to preserve abdominal organs. It is an intracellular fluid (ICF)-type solution with a low Na^+^ and high K^+^ concentration. This maintains the appropriate ion environment and helps to avoid cellular edema [22,23]. Currently, we can find many commercial preservation solutions, such as the Collins Solution, UW, HTK (histidine-tryptophan-ketoglutarate solution), Celsior, HC-A (Hypertonic citrate adenine) and IGL-1 (Institute Georges Lopez-1). Among them, there are intra- and extracellular solutions [21]. Perfusate solution contains multiple ingredients that maintain the good condition of the kidney. To prevent interstitial edema, hydroxyethyl starch is added; moreover, preservation solutions are enriched with glutathione as an antioxidant, as well as adenosine and phosphate to stimulate ATP production [24]. Schreinemachers et al. have compared three well-known preservation solutions [25]. The immediate cooling down of a graft is necessary to slow down the metabolism. According to the study, HTK solution showed the least effectiveness, and UW and PS solutions worked faster. For the appropriate washout of red blood cells, it was thought that the preservation solution should present colloid as one of the ingredients, and it should also have low viscosity. This shows the advantage of the PS over the rest. Additionally, the histological analysis of the tissue indicated that the highest edema was observed for the group washed out in HTK. This explains the fact that kidneys after HTK washout gained weight, in contrast to kidneys in PS and UW [25]. The advancement in the production of appropriate solutions was also achieved for mechanical perfusion. The comparison of UW solution and Belzer MPS solution showed that UW solution better protected DCD grafts exposed to 75 min of warm ischemia [26]. Better protection was noticeable with machine perfusion compared to simple cold static storage for both solutions. However, there was no distinction between MPS and UW when the warm ischemia time (WIT) was up to 60 min; the significant difference was stated for 75 min of WIT. The MPS solution was not effective enough, as high serum creatine values were found, and the animal survival was 25% (while for 60 min of WIT, the survival was 100%). The UW solution protected grafts more successfully. The serum creatinine values were lower than for the MPS solution, and animal survival was significantly higher—86% [26]. Presently, the most widely used preservation solution for the machine perfusion of kidneys is KPS-1. This solution is a variation of the UW solution and contains, among other ingredients, gluconate as a main anion and hydroxyethyl starch (HES), a synthetic colloid [27]. For normothermic machine perfusion (NMP), solutions must contain the ingredient that allows oxygen to be carried. For this reason, the red blood cells (RBC) and fresh frozen plasma (FFP) were used for NMP at the beginning [27,28]. However, the problem with the potential risk of infection and immune reaction of RBC led to the search for a new solution. Furthermore, we observe the red blood cells deficiency in the biobanks [29]. As an alternative to blood products, an artificial oxygen carrier was put into service. A polymerized bovine hemoglobin-based oxygen carrier (HBOC), called Hemopure, has parameters of oxygen exchange similar to RBC, is less immunogenic and can be used for NMP [30,31]. The comparison of RBC and HBOC was carried out for 6 h NMP perfusion of a kidney graft [31]. The results of blood flow, urine production, creatinine clearance and oxygen consumption were similar for both oxygen carriers [31]. A similar effect was obtained for liver perfusion [32]. This suggested that HBOC might be effectively used as a substitute for RBC for NMP.

## 5. Cold Static Storage

Cold static storage as a graft preservation method has the longest history in transplantology, and it is still the most common strategy for kidney transplants. It seems to be the most cost efficient and easiest way to provide hypothermal environments. For many years, the main goal was to create the best preservation solution to protect cells from ionic instability, edema, the toxic impact of ROS and risks. Low temperature during the hypothermic organ preservation is significant and should be established between 4 and 8 °C [33]. Unfortunately, this classic technique of cold preservation usually does not secure accurate conditions, because the temperature drops down under 2 or even 0 °C. Besides, the cooling process can run too slowly to be protective. While 4–8 °C maintains cellular metabolism at a low level, ensuring high energetic ATP savings, temperatures under 2 °C might lead to protein denaturation and damage of the cell structure [33,34]. Cold static storage is effective in short-term preservation and in conjunction with DBD and living donor organs. Since the expanded criteria for donors were approved, the classic preservation method is no longer adequate. It was found that CS-stored organs from DCD had higher rates of primary non-function of up to 9%, compared to only 1–2% in cases of donation after brain death. Similar situations occurred in delayed graft function: 22–84% (DCD) and 7–25% (DBD) [34,35]. This yields the conclusion that researchers and clinicians need to obtain better storage methods to reduce ischemia effects and increase the pool of organs available for transplantation [34].

## 6. Machine Perfusion

The history of dynamic organ preservation started in 1935, when Carrel and Lindbergh invented and described the first machine for organ perfusion [36]. Since that time, the idea has grown, and currently, different devices are produced for various techniques, starting with HMP, followed by SNMP and ending with NMP. Machine perfusion allows for the maintenance of the flow and supply of the perfusion solution with the pharmacological and biological components used as an IRI therapy (Table 2). The clinically used device for HMP contains of a chamber for ice and the cannulated organ, a pump and sensors (pressure sensor and flow sensor) and the bubble trap (Figure 2) [37].

### 6.1. Hypothermic Machine Perfusion and Pulsatile Cold Machine Perfusion

Hypothermic machine perfusion has a long history, and many investigations have been completed. Currently, transplant clinics utilizes two main types of perfusion machines: LifePort^®^ (Organ Recovery System, Itasca, Illinois) and the Waters RM3^®^ kidney perfusion system (Rochester, Minnesota) [43]. Since ischemia injury occurs during organ storage before transplantation, it is crucial to reduce it. The concept of using hypothermia is the basis for both the CS and HMP techniques (pulsatile and non-pulsatile perfusion). Hypothermia changes cellular metabolism and slows down the majority of biochemical reactions, thus decreasing energy demand. HMP additionally reduces the activity of the neutrophils and platelets during reperfusion, so the inflammatory response is decreased [7]. Cell depolarization, which is an effect of anoxia and ischemia, can be limited by pharmacological prevention and/or cooling down the graft. Special perfusion fluids have been developed. The addition of some chemical substances, such as glutathione, may control the production of free radicals due to its free radical scavenger properties. The ion imbalance during ischemia causes cellular edema. The clinically used solution maintains appropriate ion levels. Admittedly, HMP is related to higher machine cost; however, prospective economic benefits redeem it. The reduction of DGF and PNF incidents, as well as a higher percentage of one-year graft survival, reduce the total cost of hospitalization and dialysis. Machine perfusion facilitates samples collection and hence, the reliable functional and conditional control of the perfused organs. Constant flow removes toxic and metabolic products and provides good patency of the vascular bed. However, the disadvantages are cytoskeletal changes and the induction of stress proteins or the loss of cellular phospholipids [44,45,46]. Prolonged hypothermia might lead to the destruction of the organ structure [47,48]. However, HMP and its modification, pulsatile cold machine perfusion, (PP) have positive effects on kidney grafts. PP seems to be a more physiological option because it reflects better blood flow in the vessels. It has been shown that vascular endothelium reacts to mechanical stimuli, incusing flow-dependent vasoprotective genes, such as Krüppel-like Factor 2 (KLF2) that regulate the endothelial transcription of genes connected with inflammatory, innate immunity and vascular tone. The transcription is much lower during cold static storage. KLF2 positively regulates endothelial nitric oxide synthase (eNOS), consequently leading to the production of nitric oxide (NO), which is responsible for vascular conductance during reperfusion [49]. Moreover, animal models showed that there was no significant difference for the two techniques (HMP and PP) when the pulsatile mode was used. However, nonpulsatile treatment had no renoprotective effect, while pulsatile did have a good impact on a porcine graft [49]. It is speculated that PP perfusion reduces the risk of HCV transmission [50]. According to Matsuoka’s research, kidneys subjected to PP procedures had around a 10% lower rate of DGF than CS kidneys, but there was no significant difference in the case of primary nonfunction. At the same time, the DGF frequency was higher in CS grafts, while a later percentage of rejection (after 6 months and after one year) was almost at the same level between CS and PP [24]. Kwiatkowski et al. observed that the early graft function of CS and PP was similar as well, as was the rate of acute rejection (within 6 months). DGF occurred more frequently in case of CS storage, and there was more often a need for patient dialysis after transplantation (CS = 50% vs. PP = 25%) and after long term observation. The biopsy of kidneys confirmed that chronic rejection and interstitial fibrosis frequently result in cold storage grafts [50]. The meta-analysis performed by Arcos et al. indicated that there was less incidence of DGF in all type of grafts from cadaver donors in the PP group. Nevertheless, there was no significant difference between the PP and CS methods in terms of creatinine or creatinine clearance levels after the 3rd month. Furthermore, PP decreased the PNF phenomenon level, but only in reference to the DBD studies [14]. Additionally, researches on pharmacological protection during kidney perfusion are conducted [48,51]. Moser et al. proved that the administration of doxycycline (an inhibitor of matrix metalloproteinases) to the perfusion solution (KPS-1) reduced kidney injury. Markers of injury, such as NGAL concentration and LDH activity, were significantly decreased [48]. Lin et al. suggested that aldehyde dehydrogenase 2 (ALDH2) might mitigate kidney IRI [51]. They also confirmed that HMP is more efficient than CS in protecting the kidney graft. ALDH2 takes part in metabolizing aldehydes in the mitochondria and lipid peroxides, as well as resists ROS production [52]. Moreover, the investigated groups, HMP + Alda-1 (agonist) and HMP + CYA (inhibitor), have allowed for the examination of the protective effect of ALDH2. The expression of mRNA and the proteins ALDH2 were the highest after Alda-1 addition. Consequently, the lowest levels of markers of oxidative stress, injury and inflammation were observed for the HMP + Alda-1 group and the highest for the HMP + CYA group. The study indicated that the ALDH2-Akt-mTOR pathway may be a therapeutic target to protect kidneys from IRI [51].

### 6.2. Subnormothermic Machine Perfusion

The second machine-perfusion technique, subnormothermic machine perfusion (SNMP), is based on temperature settings under normothermia conditions. The temperature is between 20 and 25 °C. This is another suggestion for controlling cellular metabolism and protecting organ structures against damages arising from cold preservation [53]. Compared to NMP, lower temperatures can minimize mitochondrial injury and reduce the rate of depletion of ATP and the rate of energy changing. Unfortunately, liver transplant studies have shown that the reduction of glutathione was higher using this method, and that hepatocytes cells could not produce glutathione, probably because of the massive accumulation of cytotoxic 5-oxoproline. This may lead to metabolic acidosis and hemolytic anemia and threaten the rejection of the transplanted liver [54]. A big advantage of SNMP seems to be that there is no need to use oxygen carriers to provide gas exchange, such as the supplementation of perfusion fluid with red blood cells (RBC) or artificial substitutes. This reduces costs, as well as rising potassium levels when RBC is added [53,55,56]. During SNMP, higher pressure can be used compared to HMP, and the endothelial and vascular impartment is not observed [57]. It has been demonstrated that subnormothermic conditions might show an advantage over hypothermic storage and perfusion. Parameters such as creatinine clearance (lower in HMP) and γGT (the lowest in SNMP group) correlate positively with the kidney injury [57]. Schoop et al. subjected tissue to SNMP for 3 h after cold storage and right before reperfusion. The aim of this study was to document if the gradual heating of a graft has a positive effect on the functional parameters of the kidney. After SNMP, creatinine and urea clearance were significantly higher. It was noticed that the mitochondrial recovery was better after a short-term of SNMP treatment than in a control group and the oxygen consumption was higher. The conclusion was that controlled rewarming might have a positive impact on graft function [58]. It has been proven that gradual rewarming has a positive impact on graft viability, comparing to sudden temperature change from hypothermia to normothermia [59]. However, gradual rewarming is associated with the supplementation of oxygen carriers, which must be artificial due to the hemolysis of red blood cells used for NMP [60]. Mahboub et al. showed that the gradual rewarming of the kidney graft with HBOC after 2 h of CS improved its viability and function. The higher GFR and ultrafiltrate production had been noticed when HBOC was applied. Moreover, the reabsorption of sodium (the parameter of renal tubular function) was superior for the HBOC group compared to the control group (rewarming without oxygen carriers) [61]. One of the main problems with HMP is the time-related vascular resistance increase. It has been shown that subnormothermic conditions might be a solution to this problem, without exposing the graft to higher metabolism. Comparisons of CS and SNMP suggested that after 120 min of storage/perfusion, the vacuolization was lower in the case of SNMP, as was the necrosis. It was observed that during CS, the necrosis had been extensive, while SNMP protected the organ, and there had been only isolated necrosis in a liver study [56]. However, the protective effect of subnormothermia is still uncertain [62]. Adams et al. have compared one hour of machine perfusion after 23h of CS, using both a normothermic (37 °C) condition and with the temperature lower than normothermia (32 °C—which is not the temperature normally used in SNMP). The protective effect was observed for NMP, but not for SNMP. The creatinine clearance was lower for SNMP than the control group (without machine perfusion) and NMP. Renal damage was greater for the subnormothermic group than for the others (higher level of LDH and AST). This study suggested that only normothermic conditions are able to improve graft function after cold storage [62]. Subnormothermic and normothermic machine perfusion are proposed as recovery techniques to be utilized after cold preservation. The positive effect on liver grafts were reported in the preliminary study [63]. Ciria et al. maintained DCD grafts in a hypothermic condition and after prolonged CIT, used SNMP as a recovery method. It was shown that the rise of temperature supported graft efficiency improvement. They postulated to use devices with adjustable temperature settings (between 4 °C and 37 °C) to obtain a better condition of marginal grafts [63]. The SNMP might also reduce mitochondrial damage and free radical production [64].

### 6.3. Normothermic Machine Perfusion

The kidney outside the human body is exposed to damage associated with a lack of oxygen and nutrition. The degree of injury is associated with the time of storage before transplantation, called cold ischemic time (CIT). Since CIT is known as an independent risk factor for DGF and graft failure, it is important to reduce it. NMP, with the use of specific perfusates based on natural or artificial oxygen carriers, can provide appropriate nutrition levels and gas exchange. Kidney function can be monitored during machine perfusion when the condition is the closest to the condition of the human body. Additionally, normothermia ensures the adequate uptake of specific substances and drugs administered to perfusate fluid during machine perfusion [65]. Early studies on a short period (60 min) of NMP, right before the ECD kidney transplantation, decreased DGF. There was no incident of PNF, and the graft survival after 12 months was similar for study and control groups (CS kidneys) [66]. The results of this first human clinical study proved that NMP might be considered safe and probable. It has been shown that NMP provides better early graft and rapid tubular function [67]. Urbanellis et al. compared three techniques of kidney preservation [67]. After 30 min of warm ischemia, kidneys were perfused (HMP or NMP) or immersed in the cold preservation solution for 16 h. The renal flow and intrarenal vascular resistance were measured in real time. NMP kidneys had lower resistance results, and the blood flow progressively increased during 16 h of perfusion. The creatinine clearance was impaired in both CS and HMP compared to NMP. The fractional excretion of sodium (the marker of tubular function) was the lowest in the case of NMP. However, injury markers (NGAL and TUNEL staining) were increased only for CS kidneys, as was the histopathological damage score. This suggests that normothermic machine perfusion can protect kidney grafts from injury and improve the initial functioning of the transplant [67]. Kaths et al. compared different times of NMP applied after CS or right after warm ischemia [68]. It was shown that the flow rate reached physiological levels after 1 h of rewarming, and the intrarenal resistance decreased during NMP. The creatinine and blood urea nitrogen level (BUN) were significantly lower for the 16 h NMP group (NMP after warm ischemia without previous CS), and a smaller concentration of the serum NGAL level was decreased in groups of 16 h and 8 h in the NMP compared to control group (16 h of CS) [68]. In the pilot study of Hosgood et al., normothermic machine perfusion was used for the resuscitation of previously rejected grafts for transplantation. Research has shown that two kidneys that could not have been used because of inadequate in situ perfusion, after normothermic machine perfusion, were good enough for transplantation. The kidneys changed appearance from purple colored (signifying the damage of organ) to natural pink, the level of renal blood flow increased and a large volume of urine was produced. After this procedure, kidneys were transported in ice and transplanted. The kidneys had initial graft function and had a good level of renal function 3 months after surgery [69]. Another pilot study of 2 h of NMP of kidneys from the Eurotransplant Senior Program (ESP) confirmed the safety of this procedure. The study compared, among others, the DGF and PNF rates, creatinine levels and eGFR of the NMP kidney group, and both the ESP patient and the contralateral kidneys as a control group. DGF incidents were less frequent for the NMP group, although there were limitations of the study, such as a small sample size and no statistical differences between the study group and the control. The main conclusion was that the NMP technique is safe for the ESP group, and the kidney function after NMP was similar to that after using the standard procedure (HMP and cold storage). In the future, NMP may show the possibility of improving the storage technique with IRI therapy and the constant monitoring of relevant biomarkers of viability [2]. Studies have proven that NMP is a safe technique, and even a short period of normothermic perfusion might improve kidney function. It is important that this effect be observed before transplantation, and then it can be used to successfully resuscitate grafts. The investigation for additional graft repair methods is one of the main goals. Centers all over the world are focused on refining the NMP technology to define the ideal time of kidney normothermic perfusion and the best composition of perfusing fluid. First, solutions for NMP were based on packets of RBC and fresh frozen plasma, obtained from a local blood bank. Later, there were attempts to use a hemoglobin-based oxygen carrier (HBOC)-201. It is possible that NMP conditioning by itself can reduce IRI and at the same time, it might be a good platform for additional treatment [70,71]. The magnetic resonance imaging (MRI) used for the evaluation of NMP kidney perfusion indicated the differences between the distribution of perfusate flow inside the kidney. This suggests that the establishment of good imaging techniques might be necessary to evaluate the kidney before transplantation [72]. Still, NMP should be carefully investigated in the case of marginal donors to increase the safety of this procedure and establish one well-described protocol for transplantation centers worldwide. It seems to be crucial to substitute previously detected viability markers with imaging of actual perfusion flow and distribution inside the organ.

## 7. Novel Possibilities for the Treatment of Low-Quality Grafts and Immune-Response Modulation via NMP

Normothermic machine perfusion is the youngest technique for the storage of kidney graft and currently, the most carefully investigated. The advantage of this method were previously mentioned, and in this paragraph, we want to sum up new studies on NMP as a platform for kidney preconditioning. The greatest benefit of NMP is probably that this platform might, in the future, allow provide additional treatment for poor-quality grafts. Because of the physiological temperature and oxygen carriers, it is easier to predict cellular metabolism and drug uptake.

One of the major problems, in addition to ischemia-reperfusion injury, is that the immune response is partly downregulated by immunosuppressive drugs. However standard therapy is burden for a patient. Therefore, one of the current trends, in combination with normothermic machine perfusion, is cell-based therapy with multipotent adult progenitor cells (MAPC) and mesenchymal stromal cells (MSC) [73,74,75,76,77,78]. Both show anti-inflammatory and immunomodulatory effects and thanks to low levels of MHC II and MHC I molecules and costimulatory molecules, they are thought to be safe for the graft recipient [78]. Several studies were driven with MAPC or MSC obtained from different tissues, usually bone marrow and adipose tissue [79]. Pool et al. conducted NMP using a porcine graft while human MSCs were infused. They proved that it is possible to detect the living MSCs inside the kidney graft, but this is limited to the lumen of glomerular capillaries. There were no hemodynamics changes in the graft; however, the distribution of MSCs inside the well-perfused kidney were very diverse and hard to predict [74]. Thompson et al. described an immunomodulatory effect and a decrease of IRI markers, such as urine concentration of NGAL, and higher urine output after MAPCs infusion during human kidney NMP [77]. Besides, they noted a lower level of pro-inflammatory cytokine IL-1β and an increase in IL-10, known as an anti-inflammatory cytokine. The activity of indolamine-2, 3-dioxygenase (IDO), which modulates the effective T cells response, was also significantly higher for the NMP + MAPC group compared to the NMP control group. Similar to the results of other researchers, the staining of MAPC and kidney sections showed the increased concentration of MAPCs within the glomeruli [77]. Another study with MSCs derived from adipose (A-MSC) and bone marrow (BN-MSC) tissue showed similar results in the context of injury markers, as shown previously, such as higher levels of IL-6 and IL-8 cytokines for MSCs groups, and did not detect significant differences between the A-MSC and BM-MSC infused groups [76]. The secretion of IL-6 and IL-8 were observed only when the kidney was connected to the circuit, and the MSCs alone did not produce the mentioned interleukins. Therefore, the production of interleukin is stimulated by the injured graft. However, the origin of this product is not clearly defined; it is possible that the MSC produced interleukin, or that the kidney cells produced it upon contact with the MSCs [76]. Another cell-based therapy currently being investigated is the use of stem cell-derived extracellular vesicles (EV). EVs are secreted by cells and they take part in cellular communication because they carry small molecules like mRNA, miRNA, proteins and lipids [80]. The ability to modulate an immune response and tissue repair has been proved on AKI and CKD models of kidney injury [81]. So far, the utilization of EVs as a therapy in the context of NMP was noted for liver graft perfusion, with a satisfactory effect [82]. Kidney-derived EVs are mostly studied as novel biomarkers [80]. The possibility of EVs application during machine perfusion was studied by groups using hypothermic oxygenated machine perfusion, and they proved that even without a perfect condition of vesicles for cellular uptake (low temperature), the endocytosis occurred, and the condition of the discarded human ECD kidney significantly improved [83]. We speculate that the positive effect may be even more noticeable with NMP. Normothermic machine perfusion has been proposed as a good platform for molecular therapy to deliver small RNA fragments like microRNA antagonists [84]. MicroRNAs are non-coding oligonucleotides which, by binding to specific fragments of messenger RNA (mRNA), lead to the degradation or inhibition of the translation of that mRNA. Moreover, microRNA and other RNA-type products are small enough to be taken and filtered by the glomerulus. However, this kind of therapy is challenging because in the past, it required the delivery of small oligonucleotide encapsuled in the viral vector or small vehicles with cationic charge. However, NMP allows for the retention of physiological temperature and drug uptake; therefore, it is possible to deliver gymnotic, or naked, oligonucleotides. This method is cheaper and has less side effects. Thompson et. all used NMP to deliver an antagomir targeting mir-24-3p and recorded the positive effect on protective gene expression, such as heme oxygenase-1 and sphingosine-1-phosphate receptor 1. Besides, the tracking of the antagomir provides information about its localization in the tubular epithelial cells and endothelial cells. This confirmed the suggestion that small oligonucleotides can be used for the improvement of the condition of tubular cells, which are the most vulnerable to ischemia reperfusion injury [84].

## 8. Conclusions

The history of kidney transplantation can be dated to 1954, when the first successful transplants was carried out by American clinicians. Since that time, many changes have taken place in transplantology [85]. All this effort is supposed to lead to better organ preservation, the reduction in a ischemia-reperfusion injury and the possible extension of the time that the graft can remain outside the body. Machine perfusion was detected as a suitable method to reduce the risk of DGF and PNF, especially with DCD grafts [44,45,46,50]. Moreover, the constant control of graft function is possible during perfusion and, additionally, NMP allows for the control of the actual values of some parameters, thanks to the maintenance of physiological temperature. The technique of NMP or SNMP might also be a way for the preconditioning of the organ before transplantation by gradually increasing its temperature [63]. It is required to refine machine perfusion and elaborate repeatable rules for the procedure to create better standards for transplantology. Currently, hypothermic machine perfusion is recommended for DCD grafts in many countries. We follow this opinion, since the positive effects of HMP have been well-documented. We believe it might be the best technique for long-term organ storage, but the possibility of additional treatment should be taken into consideration to prevent distress caused by subsequent reoxygenation and rewarming. In our opinion, in the future, NMP techniques might be perfect to modify the immune system of the perfused graft if the cellular or EV therapies are proven to be safe. It would be possible to support subsequent immunosuppressive therapy at the stage of perfusion of the isolated organ. That might be extremely advantageous for ECD grafts, especially from donors already affected with immune or hormonal irregularity. Moreover, it is possible that in the future, the best kidney graft preservation method will be the combination of all three perfusion techniques, with gradual rewarming and oxygenation with the therapy “on pump.” For now, the biggest challenge is to optimize known preservation systems to make them low cost, safe and more effective. Understanding the physiopathology of IRI will allow researchers to obtain a solution containing inhibitors for enzymes and processes that might cause cell damage. All these will allow better protection of the graft and increase the pool of organs in order to increase the rate of transplantations.

## Figures and Tables

**Figure 1 biomedicines-10-01013-f001:**
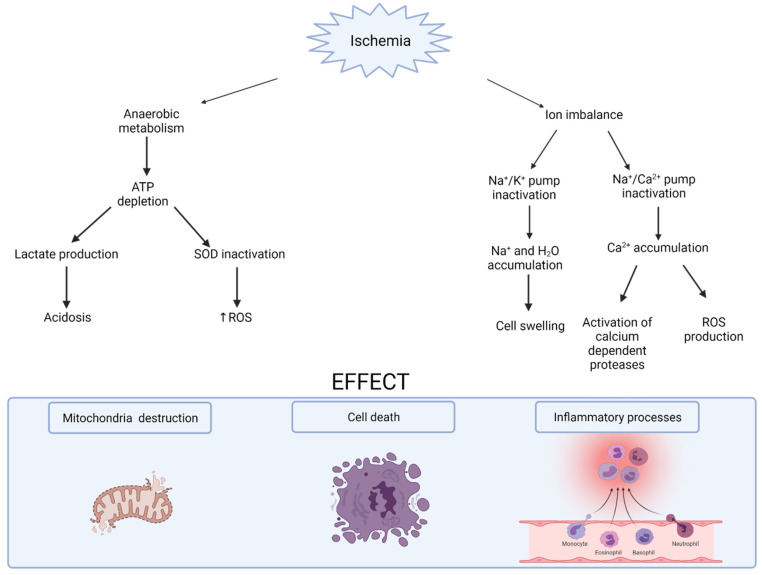
The effect of ischemia on cell metabolism. Legend: SOD—superoxide dismutase; ATP—adenosine 5′-triphosphate; ROS—reactive oxygen species.

**Figure 2 biomedicines-10-01013-f002:**
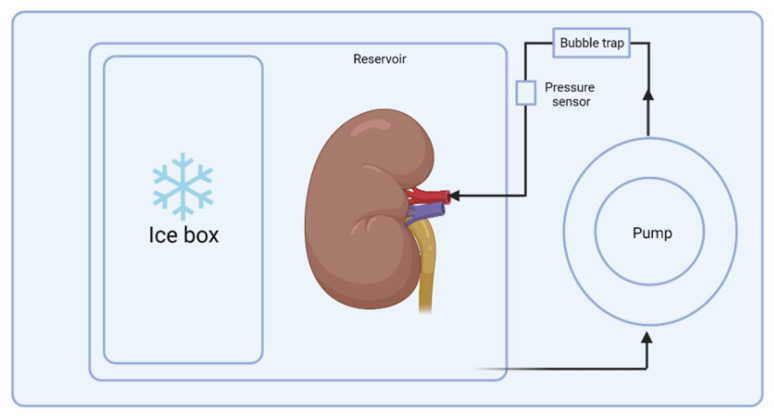
The scheme of hypothermic machine perfusion of a kidney.

**Table 1 biomedicines-10-01013-t001:** Kidney transplant and waiting patient statistics for 2018 and 2019 [13]. Those 2018 and 2019 data are based on the Global Observatory on Donation and Transplantation (GODT) data, produced by the WHO-ONT collaboration. Legend: DCD—donors after circulatory death; DBD—donors after brain death; WL—waiting list.

Country	Poland		Germany		USA		UK		Australia		Brazil	
Year	2018	2019	2018	2019	2018	2019	2018	2019	2018	2019	2018	2019
Total kidney transplants	946	983	2291	2132	22,003	24,273	3642	3649	1135	1095	5975	6298
Deceased kidney transplants	906	931	1653	1612	15,561	17,406	2608	2627	897	857	4942	5227
Actual DBD	494	503	955	932	8589	9152	1000	964	400	376	3529	3767
Actual DCD	4	1	n/d	n/d	2133	2718	619	689	154	172	n/d	n/d
Total number of patients active on the WL during whole year	2745	2747	10,616	10,325	88,595	96,830	9072	8984	1979	2027	33,201	36,371
Waiting list (Patients awaiting a transplant—only active candidates—on 31/12/201_)	1196	1165	7526	7148	60,901	60,566	5000	5030	982	1076	22,736	25,146
Patients who died while on the WL during the whole year	73	75	453	387	3934	3754	283	240	6	n/d	1402	2156

**Table 2 biomedicines-10-01013-t002:** Comparison of different machine perfusion techniques. Legend: HMP—hypothermic machine perfusion; SNMP—subnormothermic machine perfusion; NMP—normothermic machine perfusion; KPS^®^—Kidney Perfusion Solution; BELZER MPS^®^ UW—BELZER MPS^®^ solution.

	Machine Perfusion Method		
Parameter	HMP	SNMP	NMP
Temperature range	2–8 °C	20–25 °C	37 °C
Perfusion solution (examples) [22,23,29,30,31]	- KPS^®^ - BELZER MPS^®^ UW machine perfusion solution	- BELZER MPS^®^ UW machine perfusion solution - solution with oxygen carrier (natural and artificial ex. (HBOC)-201)	- Solution with oxygen carrier (natural and artificial ex. (HBOC)-201)
Equipment specifications [37,38,39,40,41,42]	- ORS (organ recovery system) LifePort - RM3 Kidney Perfusion System (Water Medical System) - Organ Assist, Groningen, the Netherlands - Waves IGL (Lissieu, France)	- Organ Assist, Groningen, The Netherlands- Modified RM3 Kidney Perfusion System (Water Medical System)	- Organ Assist (XVIVO), Groningen, the Netherlands - PerKidney^®^, San Giovanni in Persiceto, Italy - Ark system EBERS, Zaragoza, Spain
Oxygen	No/Yes (for most devices, it is possible to use oxygen)	Yes	Yes
Advantages	- reduction of cellular metabolism - reduction of ATP depletion - reduction of neutrophil and platelet activity - possibility to obtain perfusate sample for constant graft monitoring - possibility for drug administration - collection of waste products	- mitochondrial injury minimalization - decreased ATP depletion - minimalization of the vacuolization and cell necrosis - possibility of using the SNMP as a resuscitation platform for graft - possibility for drug administration and graft monitoring - collection of waste products - no compulsion to use oxygen carriers to provide gas exchange	- similar nutrition and oxygen level as in vivo condition - possibility to control graft function - adequate drug uptake when administrated - reduction of free radical mediated damage - possibility of using the NMP as a resuscitation platform for graft - collection of waste products
Disadvantages	- some cytoskeletal changes - induction of stress protein - increased ROS production after reoxygenation - loss of cellular phospholipids - the inability to accurately assess certain activities of the organ: urine production	- do not fully protect the graft from reperfusion injury - impairment of the physiological functions of the organ (in case of liver transplantation, decreased bile production)	- high level of ATP depletion - high metabolism level - high cost of a perfusion solution - risk of infection (while blood-based solution is used)

## Data Availability

Not applicable.

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
