# Peer review of "Recent Methods of Kidney Storage and Therapeutic Possibilities of Transplant Kidney"

_biomedicines, 2022, doi:10.3390/biomedicines10051013_

Round 1

Reviewer 1 Report

This is a pretty good summary of the history of storage techniques for kidneys. It spans cold static storage and progresses to perfusion machines for kidneys. However, there is very few “cutting edge” interventions reported. It is basically a summary of the organ storage techniques.  There are many types of these reviews out there and this one does not add much to what we already know and what has been reviewed elsewhere.

I recommend the authors focus on the recent discovery like the title of the manuscript, instead of covering the whole history. For example, they may review the recent papers about the normothermic machine perfusion for the studies about the method extend the preservation, reduced the IRI. There were different ways and perfusates have been reported improve it. The second part the reader may be interested in is modifying the immune character of the organ during the NMP. It may be a therapeutic strategy for future organ transplantation.

Author Response

Thank you for the review, we have found it justifiable and important for improvement of our paper. We followed the advice and we added the paragraph about novel therapies that are under investigation with the use of NMP as a platform for graft conditioning. This paragraph extended the previous short information about cellular therapy that might reduce IRI and modulate immunological environment of the graft. Besides we added some new studies about utilization of EVs and molecular modification.

Reviewer 2 Report

Overall comments:

I congratulate the authors with this basic overview of static cold storage and different machine perfusion strategies for kidney preservation. Well-written and documented and up-to-date.

Major comments:

My only major comment is the lack of clinical guidelines in this manuscript. Which perfusion technique (HMP/SNMP/NP) do you suggest in different clinical situations? But maybe this was not the aim of the manuscript.

Minor comments:

Page 1 line 24 and 25 : pharmacological and cellular therapies  : Maybe better to remove this from the abstract session because this is not the main subject of the manuscript.

Page 1 line 11 : implantation, not implementation.

Page 1, line, 18 : Better use donation after circulatory death (DCD) instead of DCDD.

Page 1, line 18 :better : Different variant of kidney machine perfusion are evaluated.

Page 3, line 117-118 : add some recent references to power this statement.

Page 5, line 191-193 : add reference.

Page 5, line 194 : remove heart-beating donors, and replace it by donation after brain death (DBD).

Page 6 : parameter Oxygen : The LifePort Kidney Transporter has since end 2021 the possibility to add oxygen. Maybe specify : Oxygen : No/yes for HMP

Page 6 table 2 : Why mentioning all the liver devices if this article is only dealing on kidney perfusion ?

Page 7, line 215 : explain more in detail the 2 types of perfusion machines. If you’re not familiar with the LifePort and RM3 this sentence is not understandable. Move line 248-252 p 8 after line 215 on page 7. Easier for the reader to understand this this sentence.

Page 8, liine253 : Explain better for the reader the Pulsatile cold machine perfusion, as a variant of HMP. Difficult to understand in the current format.

Page 9, line 330 to 336 : liver-related topic. Make it sense to mention it in this manuscript about kidney preservation?

Page 10, line 380 : NMP technique is comparable safe to HMP. I should rephrase this sentence. NMP has only be applied for kidneys as end-ischemic preservation strategy. NMP is logistically much more challenging than HMP and the risk of organs lost in case of technical problems during perfusion in NMP is much higher as compared to HMP. Also, for kidneys, no NMP device exists at this moment to realize continuous NMP. I agree with your statement in case of livers based on the continuous NP trial but most of the transplant centers are not applying NMP in their clinical liver transplant program because of the high costs, technical complexity and risk of organs lost during NMP.

Page 11, line 419 to 421. Is this statement not to specific ? Also the lack of reimbursement of this technology is one of the main reasons that transplant centers not putting kidneys on HMP or end-ischemic NMP. This will probably not change by a new preoteomic analysis of the perfusate.

Page 11, line 403 to 426 : Quite a long conclusion section. Try to summarize and add maybe a clinical guideline when to use which technique ?

Author Response

Thank you very much for your comments and advises that allowed us to improve our manuscript. As you suggested we added small comment at the conclusion part for possible clinical utilization of the mentioned techniques. Besides we have focused a little bit more on NMP and novel therapies: cell-based and molecular approach. We followed your minor comment to make it more clear and readable. We hope you will find it more interesting and accurate.   

Minor comments:

Page 1 line 24 and 25 : pharmacological and cellular therapies  : Maybe better to remove this from the abstract session because this is not the main subject of the manuscript.

We added whole paragraph about studies with NMP and possible therapies.

Page 1 line 11 : implantation, not implementation.

We changed this spell mistake.

Page 1, line, 18 : Better use donation after circulatory death (DCD) instead of DCDD.

We changed the DCDD for DCD as suggested.

Page 1, line 18 :better : Different variant of kidney machine perfusion are evaluated.

We edited that sentence. 

Page 3, line 117-118 : add some recent references to power this statement.

We added citation to prove this statement.

 Page 5, line 191-193 : add reference  

We added references. 

Page 5, line 194 : remove heart-beating donors, and replace it by donation after brain death (DBD).

We replaced the term heart-beating donors, for DBD

Page 6 : parameter Oxygen : The LifePort Kidney Transporter has since end 2021 the possibility to add oxygen. Maybe specify : Oxygen : No/yes for HMP

We clarified that information.

 Page 6 table 2 : Why mentioning all the liver devices if this article is only dealing on kidney perfusion ?

We have erased the liver machines and added kidney machines.

Page 7, line 215 : explain more in detail the 2 types of perfusion machines. If you’re not familiar with the LifePort and RM3 this sentence is not understandable.

We removed this sentence.

Move line 248-252 p 8 after line 215 on page 7. Easier for the reader to understand this this sentence.

We removed this sentence.

Page 8, liine253 : Explain better for the reader the Pulsatile cold machine perfusion, as a variant of HMP. Difficult to understand in the current format.

We explained better the pulsatile cold machine perfusion and moved this part into the text about HMP not as a separate subsection.

 Page 9, line 330 to 336 : liver-related topic. Make it sense to mention it in this manuscript about kidney preservation?

We removed various information about liver except some information about new therapies “on pump” that so far are poorly studied but in our opinion they are very promising in case of future kidney studies.

 Page 10, line 380 : NMP technique is comparable safe to HMP. I should rephrase this sentence. NMP has only be applied for kidneys as end-ischemic preservation strategy. NMP is logistically much more challenging than HMP and the risk of organs lost in case of technical problems during perfusion in NMP is much higher as compared to HMP. Also, for kidneys, no NMP device exists at this moment to realize continuous NMP. I agree with your statement in case of livers based on the continuous NP trial but most of the transplant centers are not applying NMP in their clinical liver transplant program because of the high costs, technical complexity and risk of organs lost during NMP.

We have found your question right, because it is hard to tell if the technique is equally safe as it is only small study on the specific group and as an additive proses. After second look at the mentioned paper we have changed the sentence that we believe is more suitable for the actual conclusion form the result.  

 Page 11, line 419 to 421. Is this statement not to specific ? Also the lack of reimbursement of this technology is one of the main reasons that transplant centers not putting kidneys on HMP or end-ischemic NMP. This will probably not change by a new preoteomic analysis of the perfusate.

We erased that sentence.

Page 11, line 403 to 426 : Quite a long conclusion section. Try to summarize and add maybe a clinical guideline when to use which technique ?

We shorten this section and added our subjective opinion for possible clinical usage of those techniques.

Round 2

Reviewer 1 Report

In this revision, the authors reviewed NMP with novel therapies: cell-based and molecular approaches and added a comment at the conclusion part for possible clinical utilization of the latest techniques. They concluded the advantages of different techniques and proposed a framework for future organ preservation will be the combination of 3 different techniques with gradual rewarming and therapy “on-pump”. It is reasonable. While the  NMP is currently improving, the overall benefit of NMP to improve many aspects of organ preservation is promising, making this article well positioned to impact the field of organ transplantation.

Author Response

Thank you for your comment we assure that we have corrected the mistakes.